# OpenReview forum: "Personalized Content Restriction for Large Language Models"
_TMLR — Withdrawn by Authors_

### Review · Reviewer_yRBp · 2026-05-10

**Summary Of Contributions:**

Their contributions primarily lie in the following three aspects:

1) The authors introduce Suffix Optimization (SOP), a plug-and-play method that prevents LLMs from generating personalized restricted terms by appending an optimized suffix to the original prompt, while maintaining output quality and semantic relevance.

2) The authors propose CoReBench, a new benchmark specifically designed to evaluate SOP and other representative baselines for personalized content restriction.

3) The authors evaluate SOP on several open-source LLMs and online platforms, and report improved restriction effectiveness with limited degradation in generation quality.

**Audience:**

Yes

**Audience Explanation:**

The topic is relevant to researchers working on LLM safety, controllable generation, and prompt optimization. The paper proposes a prompt-level method for suppressing user-specified restricted terms without modifying model parameters. However, the contribution is mainly about exact lexical restriction, so the interest is narrower than the paper’s broader framing of personalized content restriction.

**Broader Impact Concerns:**

The main broader impact concern is dual use. The method could support user-specific safety and privacy needs, but it could also be used for censorship or opaque lexical filtering. The paper should discuss this risk and clarify transparency expectations when such restrictions are applied.

**Claims And Evidence:**

No

**Claims Explanation:**

There are some weaknesses in the loss design:

1) Restrict Loss : The restriction loss may over-suppress legitimate words. For example, if the restricted phrase is "apple pie", the method penalizes the constituent tokens "apple" and "pie" individually, even though these words may be allowed or even necessary in a high-quality response. Thus, the loss does not precisely model phrase-level avoidance.

2) Quality Loss : The quality loss compares the suffix-conditioned output $\tilde{y}$ with the original output $y=f(x)$. However, since the original answer may contain the restricted terms, this objective can directly conflict with the restriction loss. Moreover, it is not clear that minimizing the discrepancy between $\tilde{y}$ and the original unrestricted output $y$ is an appropriate measure of output quality under restriction. In particular, preserving quality in this setting should mean producing a fluent and coherent restricted-term-free alternative, rather than reproducing the original unrestricted response.

3) Semantic Loss : The semantic loss uses the cosine similarity between the embeddings of the original prompt $x$ and the restricted-term-avoiding output $\tilde{y}$ to preserve semantic relevance. However, prompt-output embedding similarity is not a reliable measure of semantic relevance, since a correct response need not be embedding-similar to the input prompt, especially in question-answering or instruction-following settings.

4) Missing Objective: The current objective does not explicitly reward the intuitively desired behavior: replacing a restricted expression with a fluent, semantically equivalent, and allowed alternative. For example, if "apple pie" is restricted, a good response might use a phrase such as "a classic dessert made with apples" instead. It is unclear whether either the quality loss or the semantic loss meaningfully rewards this type of restricted-term-free paraphrase.

Also, the experiment section includes weakness as follows:

1) The benchmark is synthetic and filtered to elicit restricted terms, so generalization to natural user prompts is unclear.

2) The baselines are weak; stronger practical baselines such as filter/regenerate, rewrite, post-processing replacement, or token/logit bias should be included.

3) The paper reports $R_{\mathrm{qua}}$ as an LLM-judge-based quality score, but this is only a coarse proxy for output quality. It does not directly evaluate whether the restricted-term-free response preserves the original meaning, specificity, or completeness. A model may avoid the exact restricted terms by producing a more generic or less informative answer, while still receiving a reasonable quality score.

4) Furthermore, since SOP is designed to balance restriction, quality, and semantic relevance, the paper is expected to provide a more systematic trade-off analysis between restriction-rate improvement and degradation in quality, relevance, and information preservation.

**Requested Changes:**

The authors should address the main weaknesses discussed above, especially by narrowing the claims to exact restricted-term suppression, better justifying or revising the loss design, adding stronger practical baselines, and providing more realistic generalization and failure analyses. These changes are important for supporting the paper’s broader claims.

---

> ### Author Response · Authors · 2026-05-30
>
> ## **Response to Reviewer yRBp**
>
> We thank the reviewer for the detailed comments. The comments suggest that the task scope and the role of each objective should be clarified more explicitly. We will revise the paper to better emphasize that SOP targets **personalized restricted-word control**, rather than general semantic rewriting or open-ended style transfer.
>
> ## General response to loss design concerns
>
> The concern appears to stem from a different interpretation of the task objective. SOP is not designed as a general semantic rewriting method. It targets **personalized restricted-word control**, where the goal is to suppress arbitrary user-specified terms while maintaining useful generation. Under this setting, the restriction, quality, and semantic losses are intentionally designed to play complementary roles.
>
> This design is already supported by our ablation study in Appendix Table 5. The results show a clear pattern: using the restriction loss alone improves suppression but harms quality; removing the restriction loss improves quality but fails to enforce the restriction; and the full objective achieves the best overall trade-off across 3, 6, and 9 restricted terms. Therefore, the losses are not redundant or inconsistent. They encode the core trade-off of the task. We will revise the paper to highlight this ablation analysis more prominently in the main text.
>
> ---
>
> **Q1.** Restriction loss and phrase-level avoidance
>
> **A1.** The token-level restriction loss is a deliberate design choice rather than an oversight. Our task is personalized restricted-word control, where users may specify arbitrary words, phrases, names, entities, or sensitive expressions that should be avoided in generated outputs. In this setting, restricting constituent tokens provides a conservative but effective safety margin, especially when restricted expressions can appear in partial, inflected, reordered, or contextually embedded forms.
>
> Phrase-level modeling is useful when the restricted unit is always a fixed phrase, but it is less reliable when the target expression may appear partially or in flexible surface forms. In contrast, our token-level objective directly aligns with the main evaluation metric, which measures whether restricted terms appear in the final output. We will clarify this design motivation in the revision.
>
> ---
>
> **Q2.** Quality loss and conflict with restricted terms
>
> **A2.** The quality loss is included to prevent the optimized suffix from finding trivial solutions, such as forcing refusal, degeneration, or irrelevant outputs. Its role is not to copy every token in the original response, but to preserve general fluency, informativeness, and task relevance while the restriction loss suppresses prohibited terms.
>
> This design reflects the central trade-off of our task: the model should avoid restricted words, but it should not collapse into generic or low-quality responses. This is empirically supported by Appendix Table 5. When the restriction loss is used without the full objective, restriction can improve but quality drops substantially. The full objective provides a better restriction-quality balance, showing that the quality loss is necessary rather than contradictory.
>
> ---
>
> **Q3.** Semantic loss and semantic relevance measurement
>
> **A3.** The semantic loss is intended as a lightweight regularizer, not as a complete measure of answer correctness. Its purpose is to reduce behavioral drift introduced by suffix optimization. Since the optimized suffix can strongly affect model behavior, this regularizer helps prevent the model from producing fluent but off-topic or overly generic responses.
>
> We do not rely on this loss as the sole evidence of semantic preservation. The final behavior is evaluated using restriction success and response quality metrics, and the contribution of the loss components is further supported by the ablation results in Appendix Table 5. We will revise the paper to make this role clearer.
>
> ---

---

> > ### Author Response · Authors · 2026-05-30
> >
> > **Q4.** Missing objective for fluent restricted-term-free alternatives
> >
> > **A4.** We thank the reviewer for the suggestion. We would like to clarify that SOP is designed for **restricted-word control**, not for full replacement-aware rewriting. The primary goal is to prevent user-specified restricted terms from appearing while preserving useful generation. In many personalized restriction scenarios, such as avoiding private names, sensitive entities, or user-defined prohibited expressions, the key requirement is whether the restricted term is leaked, rather than whether the model always finds an optimal substitute expression.
> >
> > That said, replacement-aware generation is an interesting extension. We will clarify this scope in the revision and add qualitative examples showing different behaviors, including successful avoidance with natural paraphrasing, avoidance with information omission, and failure cases. This will better characterize what SOP currently achieves without overstating it as a full semantic rewriting method.
> >
> > ---
> >
> > **Q5.** Synthetic benchmark and generalization to natural prompts
> >
> > **A5.** CoReBench is intentionally constructed as a controlled benchmark for restricted-word control. This design is necessary because, if the model would not generate the restricted term in the first place, restriction success becomes trivial and difficult to measure. By using prompts that are likely to elicit restricted terms, CoReBench creates a meaningful stress test for whether a method can suppress specified content while preserving output quality.
> >
> > Therefore, the synthetic nature of CoReBench is not a weakness of the evaluation, but part of its purpose: it provides controllable, measurable, and repeatable test cases for personalized content restriction. This is similar to safety benchmarks that intentionally elicit risky or undesired behavior to evaluate whether safeguards are effective. We will revise the paper to make this benchmark design clearer.
> >
> > ---
> >
> > **Q6.** Stronger practical baselines
> >
> > **A6.** Our current baselines focus on prompt-level methods because SOP is also a prompt-level control method and does not require parameter updates at deployment time. This makes system-prefix and system-suffix prompting the most directly comparable baselines. We agree that additional practical baselines can further strengthen the empirical comparison. In the revision, we will include or discuss baselines under different deployment assumptions, such as filter-and-regenerate, rewrite-after-generation, post-processing replacement, and constrained decoding when logit access is available.
> >
> > We will clarify that these methods are not all directly comparable. For example, constrained decoding requires decoding-time access, while rewrite-after-generation introduces additional model calls, latency, and potential semantic drift. SOP provides a compact prompt-level controller that can be reused after optimization, which gives it a different practical trade-off.

---

> > ### Comment · Reviewer_yRBp · 2026-06-07
> > **Response**
> >
> > Thank you for the clarification. I agree that narrowing the scope to exact restricted-word control makes the task definition more precise. However, this clarification also substantially weakens the practical significance of the work. If SOP is not intended to provide semantic-preserving restricted-term-free alternatives, but only to suppress specified surface terms, then it is unclear to me whether the resulting problem formulation captures a sufficiently important real-world need. Under this narrowed scope, the paper should directly justify why SOP is practically preferable to much simpler lexical-control pipelines, such as detection plus regeneration, rewriting, post-processing replacement, or token/logit-bias methods when available. Without such justification, the revised framing may be clearer, but it also makes the contribution appear much less practically meaningful than claimed.

---

### Review · Reviewer_TFWB · 2026-05-13

**Summary Of Contributions:**

This work studies personalized content restriction for deployed LLMs without modifying model parameters. It proposes Suffix Optimization (SOP), a prompt-based method that learns a universal suffix to suppress user-specified restricted terms while preserving response quality. The paper also introduces CoReBench, a benchmark for evaluating personalized content restriction, and demonstrates that SOP outperforms other baselines across several open source LLMs. In general, strengths of this work includes the practical and lightweight formulation, reasonable empirical evaluation, and the introduction of a new benchmark. Weaknesses include the limited semantic guarantees, benchmark scale and the diversity, which may not fully capture more realistic or semantically complex restriction scenarios. Besides, the reliance on transferability from surrogate open source models to black-box business models also needs to be clarified.

**Audience:**

Yes

**Audience Explanation:**

The paper studies a practical and increasingly relevant problem of personalized content restriction for deployed LLMs, which is likely to interest researchers working on LLM safety, controllability, and prompt optimization.

**Broader Impact Concerns:**

None.

**Claims And Evidence:**

No

**Claims Explanation:**

This paper provides empirical evidence across multiple models and several ablation studies, and the reported results generally support the effectiveness of SOP under the proposed CoReBench setting. However, the current evidence is not sufficient to support some of the claims regarding robustness, generalization, and practical applicability in realistic personalized safety scenarios.
To be more specific,
1. The proposed CoReBench benchmark in Section 5 is relatively limited in both scale and diversity (only 400 prompts and 80 restricted terms), which may not adequately reflect realistic personalized content restriction settings with more complex or dynamic constraints.
2. The evaluation mainly focuses on lexical-level restriction (i.e., avoiding specific words or phrases). I wonder how SOP would perform against semantically equivalent paraphrases, synonym substitutions, or indirect references to restricted concepts? These are also common failure cases in practical safety systems. In addition, the quality evaluation relies heavily on GPT-based LLM-as-a-judge scoring. Would more detailed qualitative error analysis provide stronger evidence regarding fluency, semantic preservation, or unintended behavioral introduced by the optimized suffixes?
3. I find the writing of the transferability and online platform experiments in Section 6.3 is somewhat unclear and not fully convincing. In Section 2, the paper motivates the problem setting by emphasizing the alignment issue and API-only access limitations. However, the proposed SOP method still requires access to logits during the loss computation such as equation (2) and (4) instead of only API access scenario, which should be clarified more explicitly. Besides, the transferability experiments are conducted primarily across relatively similar open source models. Would the optimized suffixes remain effective on substantially different proprietary models that use different alignment strategies? And do the online platform experiments truly demonstrate robustness to black-box deployment settings, or just transferability across deployment environments of related models?

**Requested Changes:**

1. Expand the evaluation beyond the current CoReBench setting by including more diverse and realistic personalized restriction scenarios. The current benchmark size and diversity are somewhat limited for supporting broad generalization claims.
2. Provide additional evaluation on semantic-level restriction robustness. In particular, I would like to see experiments involving paraphrases, synonym substitutions, indirect references, or semantically equivalent expressions of restricted concepts, since the current evaluation mainly focuses on lexical-level suppression.
3. Strengthen the transferability evaluation by testing on more substantially different proprietary or black-box models, rather than primarily across relatively similar open source models. This would better support the claims regarding robustness and practical deployment generalization.
4. Some typos need to be revised: (1) “charactor.ai” should be “character.ai” in Section 3.2; (2) the benchmark name is inconsistently written as both “CoReBench” and “CoreBench” (in Appendix B).

---

> ### Author Response · Authors · 2026-05-30
>
> ## **Response to Reviewer TFWB**
>
> ---
>
> We thank the reviewer for the thoughtful and constructive comments. We appreciate the recognition that the paper studies a practical problem relevant to LLM safety, controllability, and prompt optimization. We also agree that the current version should better qualify the scope of our claims, especially regarding robustness, generalization, and practical applicability in API-only settings.
> Below we address each concern and describe the planned revisions.
>
> **Q1.**  Limited scale and diversity of CoReBench
>
> **A1.**  Our CoReBench focuses on eight deliberately selected, practical, and non-political categories, and it is **explicitly extensible**. Additional restricted terms or use-case categories can be incorporated following our protocol.  Although CoReBench contains 400 prompts and 80 restricted terms, this scale is fully aligned with established benchmarks for other safety-related tasks. For example, **AgentDojo** includes 629 security test cases, and **HEx-PHI** contains 330 harmful instructions (30×11 categories). These datasets are widely adopted not because of size, but because they consist of **carefully curated, high-impact scenarios**—a design philosophy we follow.
>
>
> **Q2.**   Semantic-level restriction robustness
>
> **A2.**  We thank the reviewer for this important comment. We agree that our current evaluation mainly studies lexical-level restriction, i.e., suppressing exact user-specified words or phrases, rather than full semantic concept suppression.
>
> We agree that semantic variants such as synonyms, paraphrases, and indirect references are important. We will add a semantic robustness analysis by testing SOP on such variants. We will evaluate both Exact-term SOP, where optimization uses only the original restricted terms, and Expanded-set SOP, where synonyms and paraphrases are included in the restriction set.
>
>
> **Q3.**   Quality evaluation and qualitative error analysis
>
> **A3.**  We thank the reviewer for this helpful suggestion. Our current evaluation follows common practice in recent LLM studies by using GPT-based LLM-as-a-judge to assess response quality. In addition, the main restriction success rate is computed by deterministic keyword matching, so the core suppression metric does not rely on GPT judgment. We also welcome the reviewer’s suggestions on additional evaluation protocols or quality metrics, and we will incorporate them where feasible in the version.
>
>
> **Q4.**   Clarification of API-only and black-box deployment claims
>
> **A4.** We thank the reviewer for raising this point. We will revise the manuscript to more clearly distinguish between the suffix optimization stage and the suffix deployment stage.
> A key advantage of SOP is that it separates optimization from deployment. During optimization, SOP learns a compact suffix using an accessible model, where logits or gradients can be used to optimize the restriction and quality objectives. Once learned, however, the suffix is a plain text prompt component and can be directly appended to user inputs without changing model parameters, modifying the decoding process, or requiring any access to the target model internals.
> Therefore, our API-only discussion refers to this deployment property. In practical settings, SOP can be optimized on a surrogate or local model and then transferred to API-only models as a plug-and-play prompt-level controller. This makes SOP different from parameter-editing or fine-tuning based methods, which require direct modification of the deployed model.
>
>
> **Q5.**   Transferability and online platform experiments
>
> **A5.**  We thank the reviewer for raising this point. Our transferability and online-platform experiments are intended to show the practical reuse potential of SOP suffixes beyond the source model. The results indicate that a suffix optimized on one open-source model can transfer to other models, particularly when the source and target models have comparable scale and instruction-following behavior. This suggests that SOP does not simply overfit to one model, but can capture restriction-oriented prompt patterns shared across related LLMs.
>
> The online-platform results further show that, once optimized on an accessible surrogate model, the suffix can be deployed through a standard prompt-only interface without accessing target-model parameters.  Importantly, even partial transfer is practically valuable, because SOP requires no target-model fine-tuning or internal access at deployment time. With stronger surrogate models and larger-scale optimization, we expect this transferability to be further improved.
>
>
>
> **Q6.**   Typos and naming consistency
>
> **A6.**  We will correct the typo in Section 3.2 by changing “charactor.ai” to “character.ai”. We will also make the benchmark name consistent throughout the paper by using “CoReBench” everywhere, including Appendix B.

---

### Review · Reviewer_aWSd · 2026-05-18

**Summary Of Contributions:**

This paper studies the problem of preventing LLMs from generating user-specified restricted words through prompt engineering. The authors propose Suffix Optimization (SOP) that injects an optimized sequence of tokens to the end of the prompt to reduce the likelihood of restricted terms appearing in the output. They also introduce CoReBench, a benchmark designed to evaluate personalized content restriction methods.

Strength:
* The proposed method is lightweight and straightforward to implement. Since the trained SOP only requires prompt-level modification, it can also be applied to black-box APIs where model parameters and gradients are not accessible.
* The empirical results show that SOP achieves strong performance, and the generalization analysis provides useful insights.
* The proposed benchmark is useful for evaluating personalized content restriction and can also provide a valuable testbed for studying jailbreak robustness under customized output constraints.

**Audience:**

Yes

**Audience Explanation:**

This method is interesting where it focus on the prompt level and may also extend to close-sourced models.

**Broader Impact Concerns:**

There's no broader impact concerns about this submission.

**Claims And Evidence:**

No

**Claims Explanation:**

The claims are mostly supported by clear experimental evidence. The authors evaluate SOP on CoReBench under different numbers of restricted terms. The evaluated models are varing across different structures and sizes. The main comparison includes no restriction, system-prefix prompting, and system-suffix prompting, and the results show that SOP generally achieves higher restriction rates while keeping a comparable quality score. The paper also includes additional analysis on SOP.

However, there are still several weaknesses.
* The algorithmic description is unclear and would benefit from further clarification. For example, it is not clear what “selected replacement” exactly refers to in Section 4.2. Does it represent a permutation over all selected candidates? In Algorithm 1, the meaning of the selection batch $B$ is also confusing. While in Line 7 of the algorithm box, the index $j$ appears without a clear definition. If $j$ is sampled randomly, the authors should explain how the robustness of the method is supported under such randomness.

* The authors present their method as an extension of GCG, but GCG is not included as a baseline in the empirical comparison. This makes it difficult to assess whether the proposed method provides a clear improvement over the method it builds upon. In addition, other prompt-level optimization methods, such as AutoDAN, are also absent from the baseline comparison.

**Requested Changes:**

The authors need to clarify the details in the algorithm and add relative baselines in the evaluation.

---

> ### Author Response · Authors · 2026-05-30
>
> ## **Response to Reviewer aWSd**
>
> We thank the reviewer for the careful reading and constructive comments. We appreciate the positive feedback on the lightweight design of SOP, the empirical results, the generalization analysis, and the usefulness of CoReBench. We also agree that the current algorithmic description should be clarified and that the empirical comparison can be strengthened by adding prompt-optimization baselines such as GCG and AutoDAN. We will revise the paper accordingly.
>
> ---
>
> **Q1.** Clarification of the algorithmic description
>
> **A1.** Thank you for pointing out the ambiguity in Section 4.2 and Algorithm 1. We agree that the current description of “selected replacement”, the selection batch \(B\), and the index \(j\) is not sufficiently clear.
>
> In our algorithm, a “selected replacement” does **not** refer to a permutation over all selected candidates. Instead, it refers to a **single-coordinate token substitution** for the suffix. Specifically, for each suffix position \(j \in \{1,\ldots,d\}\), we first compute a candidate set \(X_j\), which contains the top-\(k\) token replacements according to the negative gradient of the total loss. Then, in each optimization iteration, we construct a selection batch of \(B\) candidate suffixes. Each candidate suffix differs from the current suffix at only one position.
>
> Regarding the randomness in the sampling step, the randomness is only used to propose candidate substitutions. The final update is selected by minimizing the objective over the \(B\) proposed candidates. Therefore, the method does not rely on a single random replacement. In addition, our ablation study on the greedy search width \(B\) and the replacement size \(k\) shows that SOP is stable under different search configurations. To further address this concern, we will add more implementation details, including the sampling procedure, random seed settings, and, if space permits, mean and standard deviation over multiple runs.
>
> **Q2.** On the absence of GCG and AutoDAN baselines
>
> **A2.** We agree with the reviewer that the relationship between SOP and GCG should be better reflected in the empirical comparison. Our intention was to present SOP as an adaptation of the GCG-style discrete coordinate optimization framework to the personalized content restriction setting, rather than as a completely independent search algorithm. However, we agree that including an explicit GCG-style baseline would make the contribution clearer.
>
> We also appreciate the suggestion to compare with other prompt-level optimization methods such as AutoDAN. AutoDAN is originally designed to generate stealthy jailbreak prompts rather than content-restriction suffixes, so it cannot be directly used without modifying its objective. We will add a discussion include an empirical comparison with prompt optimization baseline in the revision.

---

### Author Response · Authors · 2026-07-06

Dear Action Editor,

Thank you for handling our submission. After carefully considering the reviews, we have decided to withdraw our submission from TMLR. We plan to substantially revise the manuscript before submitting it to another venue.

We appreciate the reviewers’ and your time and feedback.

Best regards,
Authors

---

### Note · Authors · 2026-07-06

I have read and agree with the venue's withdrawal policy on behalf of myself and my co-authors.